# Can Biological Control Agents Reduce Multiple Fungal Infections Causing Decline of Milkwort in Ornamental Nursery?

**DOI:** 10.3390/plants9121682

**Published:** 2020-12-01

**Authors:** Dalia Aiello, Alessandro Vitale, Giancarlo Perrone, Matilde Tessitori, Giancarlo Polizzi

**Affiliations:** 1Dipartimento di Agricoltura, Alimentazione e Ambiente, sez. Patologia vegetale, University of Catania, Via S. Sofia 100, 95123 Catania, Italy; dalia.aiello@unict.it (D.A.); mtessitori@unict.it (M.T.); gpolizzi@unict.it (G.P.); 2Istituto di Scienze delle Produzioni Alimentari, CNR, Via Amendola 122/O, 70126 Bari, Italy; giancarlo.perrone@ispa.cnr.it

**Keywords:** *Calonectria*, *Fusarium* and *Rhizoctonia* species, disease control, fungicides, microbiological formulates

## Abstract

This research evaluates biological control agents (BCAs) and fungicide alone and in combination for the management of decline caused by multiple fungi on milkwort (*Polygala myrtifolia*). Four experiments were performed in a greenhouse within a nursery located in Catania province (southern Italy). The activity of fungicides and biological control agents was evaluated by calculating the plant mortality (%) and recovery frequency (%) of different fungi associated with symptomatic tissue. Comprehensively, boscalid + pyraclostrobin and fosetyl-Al showed the best results in managing disease complex on milkwort. Biological control agents provided, on average, the lowest performances; nevertheless, in most cases, they were able to significantly reduce multiple infections and sometimes when combined with fungicide enhanced the effectiveness. The molecular analysis of 86 isolates obtained from symptomatic tissue allowed to identify the fungi involved in the disease as *Calonectria*
*pauciramosa*, *C*. *pseudomexicana*, *Fusarium*
*oxysporum*, *Neocosmospora solani* (syn. *F*. *solani*) and binucleate *Rhizoctonia* AG-R. *Calonectria*
*pseudomexicana* never reported on milkwort and in Europe was inoculated on *P. myrtifolia* potted healthy cuttings and produced crown and root rot after 40 days. Our findings represent the first worldwide report about disease complex of milkwort caused by several fungi (*Calonectria* spp., *Fusarium* spp. and binucleate *Rhizoctonia*) and on the effects of integrated control strategies to manage this disease in the nursery.

## 1. Introduction

Milkwort (*Polygala myrtifolia* L.) is a widespread, evergreen shrub belonging to Polygalaceae family and native to the South African regions. Generally, the use of this ornamental crop is concentrated in open gardens and parks in southern Italy, whereas it is mainly cultivated as a potted plant in northern Italy. This species can be affected by several fungal diseases. Among these, crown and root rot caused by *Calonectria pauciramosa* C.L. Schoch & Crous (*= Cylindrocladium pauciramosum* C.L. Schoch & Crous) [1], web blight, crown and root rot caused by binucleate *Rhizoctonia* (BNR) AG-R (teleomorph: *Ceratobasidium* Rogers) [2] and decline and crown and root rot caused by *Fusarium oxysporum* Schltdl.: Fr. and *Neocosmospora solani* (Mart.) L. Lombard & Crous (sin. *Fusarium solani* (Mart.) Sacc. [3] were reported in Italy. In Sicily (southern Italy), *Calonectria* species are the primary pathogens and cause heavy losses on milkwort in nursery. Diseased plants show wilting, stunting, chlorosis or loss of foliage and the rotting of the basal stem as well as crown and root rot. *Calonectria* spp. are pathogens of a broad range of ornamental and forestry crops and widespread in Italy [4,5,6,7,8,9,10,11,12,13,14,15,16,17,18]. The management of Calonectria infections in nursery should involve the development of integrated control strategies aimed at reducing both the level of the primary inoculum in contaminated soil or substrate used for cultivation [19,20,21] and the rate of infection [22,23,24,25]. Although the use of some fungicides such as benzimidazoles and sterol demethylation inhibitors is effective against *Calonectria* species, their use should be limited due to the selection of resistant isolates [26,27,28]. Besides, some biological control agents (BCAs) are effective to reduce Calonectria infections [23,29,30]. However, based on our results on Myrtaceae and Sapindaceae ornamental species, BCAs applied alone were less effective than fungicides or combined treatments [23]. Moreover, the efficacy of BCAs is variable depending on different factors such as the host species, the application modes and timing of biological agents, the fungal species and isolate and the commercial formulations used [30,31,32]. Thus, an effective integrated control strategy of Calonectria diseases in nursery can be achieved by testing both different combination of BCAs and fungicides and different modes and times of application.

Therefore, the aims of this research were to evaluate BCAs and fungicides alone and in combination to management of decline of milkwort in nursery and identify the fungal species involved in the disease.

## 2. Results and Discussion

### 2.1. Effectiveness of Treatments

Data on treatment efficacy and relative to four experiments (I, II, III and IV) performed under conditions of both artificial and natural disease pressure are reported in Table 1, Table 2, Table 3 and Table 4. Nevertheless, it was not possible to relate plant mortality data with the relative recovery frequency of the single fungal pathogen retrieved.

### 2.2. Experiment I

Results from experiment I were carried out under artificial Calonectria disease pressure and are reported in Table 1. Based on these results, it was found that plant mortality is due not only to inoculated *C. pauciramosa* but also to *Fusarium* spp. natural infections as shown in the recovery frequency column. However, almost all treatments significantly reduced milkwort seedlings’ mortality values if compared with those of untreated control. Commercial formulate containing boscalid + pyraclostrobin was the most effective treatment for controlling mixed fungal infections in milkwort. Moreover, this latter finding was partially confirmed by relative lowest *C. pauciramosa* recovery frequencies (although not significant data). Also, fosetyl-Al applied alone consistently reduced seedlings mortality when compared with most of the remaining treatments. Although with lower performances, tested BCAs, applied alone or in combinations with fosetyl-Al, significantly reduced infections of collar and root rot complex caused by *C. pauciramosa* and *Fusarium* spp. except for those of milkwort treated with *Streptomyces* K61 and *Bacillus amyloliquefaciens* D747 + fosetyl-Al combination, respectively. Conversely, data regarding fungal recovery among tested treatments did not always significantly differ among them and from untreated control (Table 1).

### 2.3. Experiment II

In experiment II performed under artificial *C. pauciramosa* disease pressure, commercial fungicide boscalid + pyraclostrobin mixture, when applied alone or in combination with tested BCA formulates always, provided a significant reduction of multiple disease infections if compared to those of untreated control and treated with BCAs alone (Table 2). However, all boscalid + pyraclostrobin-based treatments showed averagely the same performances among them in disease complex management. These findings are in full accordance with the data obtained in the previous experiment and with those recently reported by Cinquerrui et al. [23]. It is worth noting that the application of Streptomyces K61 and *T. atroviride* & *T. asperellum*, in this disease pressure context, resulted in a statistically significant decrease of plant mortality on their own, and observed data were not statistically different from the relative combination of these biocontrol agents with boscalid + pyraclostrobin. However, it was not possible once again to observe significant recovery frequency differences for involved fungal genera (*Calonectria, Fusarium* and *Rhizoctonia*) in crown and root infections among all tested treatments. Nevertheless, *Calonectria* recovery values from symptomatic tissues of milkwort treated with boscalid + pyraclostrobin were averagely the lowest among treatments thus partially confirming data of previous experiment. Otherwise, data regarding the mean recovery frequency of *Fusarium* spp. from infected tissues was very variable among tested treatments whereas BNR colonies were on average recovered with the lowest frequency among all three fungal genera. Comprehensively, better agronomic parameters (shoot number, length and weight) were almost better in all treated milkwort seedlings if compared to those of untreated ones although it was not always possible to detect significant differences among treatments (Table 2).

### 2.4. Experiment III

In experiment III, performed under natural disease pressure conditions, fosetyl-Al, applied alone or in combination with BCAs, showed the best performances (significant data) among all treatments in being able to significantly reduce plant mortality caused by crown and root rot disease complex (Table 3). These data are in agreement with those recovered in experiment I. Otherwise, thiophanate-methyl and propamocarb + fosetyl-Al mixture revealed ineffective in significantly reducing mixed infections caused by *Calonectria* spp., *Fusarium* spp. and BNR fungi. The breakdown of efficacy of thiophanate-methyl & prochloraz (and in general for MBC fungicides) could be probably related to widespread occurrence of fungal resistance phenomena [26,27,28] whereas the failure of propamocarb + fosetyl-Al mixture in controlling Calonectria infections alone was also previously reported by Aiello et al. [22]. Although *Calonectria* and *Fusarium* recovery frequencies detected in treated plots were always numerically lower than those detected in controls once again the data were not significantly different. *Rhizoctonia* recovery frequency from infected tissues of milkwort seedlings was always lesser than other fungal pathogens and very variable among all treatments. Regardless of agronomic parameters, no significant differences were detected among tested treatments (Table 3).

### 2.5. Experiment IV

In experiment IV, scheduled application of a boscalid + pyraclostrobin (Signum™) mixture and fosetyl-Al alone and combined in alternation with BCAs was effective in controlling attacks of crown and rot disease complex incited by *Calonectria, Fusarium* and *Rhizoctonia* species, being always significantly different from untreated control (Table 4). Although without significant differences, all treatments including scheduled fungicide applications combined with BCAs employment, provided a good reduction of infection amount due to *C. pauciramosa* (see fungal recovery data) and not always for *Fusarium* spp. Moreover, *Rhizoctonia* spp. was occasionally recovered from infected tissues. Overall, standard fungicide program (alternations of thiophanate methyl with prochloraz) showed the worst performances in managing mixed fungal infections in the nursery (Table 4). This latter finding should be probably related to spread resistance phaenomena to methyl benzimidazole carbammates (MBCs) and decreased sensitivity to prochloraz reported in south Italy [22,26,27,28].

### 2.6. Recovery Frequency and Identification of Isolates

According to the morphological and molecular analysis, 86 representative isolates single-conidium recovery from symptomatic tissue were identified as follows: 31 *Fusarium* species (12 isolates belong to *F. solani* species complex and 19 to *F*. *oxysporum* species complex), 45 *Calonectria* spp. (named with acronym CP followed by number) and 10 binucleate *Rhizoctonia* AG-R.

Translation elongation factor 1-alpha (EF-1α) gene (TEF) sequences of one representative isolate for both *Fusarium* species showed 100% and 99.8% homology with a strain of *F*. *oxysporum* (GenBank accession no. LS420072) and a strain of *F*. *solani* (GenBank accession no. CBS131775), respectively. *F*. *solani* and *F*. *oxysporum* were already reported by Vitullo et al. [3] and cause crown and root rot in this species. According to the new taxonomic classification, *F*. *solani* is a name given to a complex “*Fusarium solani* Species Complex” (FSSC) of over 45 morphologically cryptic species è [33] and a recent study showed that FSSC is actually another genus of the Nectriaceae family named *Neocosmospora* [34]. Also *F*. *oxysporum* is a species complex (FOSC) with different species and *formae specialis*. The FOSC includes soil-borne pathogens responsible for vascular wilt, cankers, rot and damping-off of a wide range of agronomical and horticulturally important crops [35] and widely spread in Sicily, Italy [36,37,38,39].

Analysis of the internal transcribed sequence spacer region of rDNA and 5.8S (ITS) revealed that the isolates of binucleate *Rhizoctonia* belonged to binucleate *Rhizoctonia* AG-R. One representative isolate of binucleate *Rhizoctonia* showed a sequence similarity of 99% with an AG-R isolate from Genbank (accession no. JX514382). Binucleate *Rhizoctonia* AG-R was recovered with lower frequency compared to the other fungi but still it showed a role in the disease complex of milkwort confirming its ability to cause crown and rot root as reported by Aiello et al. [2].

The identification of the *Calonectria* strains is evident from the tree shown in Figure 1, in which the phylogenetic tree of the *Calonectria* species is closely related to the 45 strains clearly evidenced, with a high bootstrap, that 43 strains belong in the clade of *C. pauciramosa* and two strains in the clade of *C. pseudomexicana* L. Lombard, G. Polizzi & Crous. Among *Calonectria* species, *C. pauciramosa* was already reported on milkwort by Polizzi & Crous [1], while *C*. *pseudomexicana* was recovered for the first time from this species. The isolates of *C*. *pseudomexicana* (CP7 and CP14) were pathogenic to the inoculated cuttings of milkwort and produced crown and root rot after 40 days. All plants show wilting, basal stem rot and crown and root rot with 100% of disease incidence while plant mortality was 64% and 71% for CP7 and CP14, respectively. The pathogen was re-isolated from the artificially inoculated plants and were identified as previously described, fulfilling Koch’s postulates. No symptoms were observed on control plants. This work shows the pathogenicity and the presence of *C*. *pseudomexicana* in Italy for the first time. This species was first described in Tunisia from *Callistemon* sp. by Lombard et al. [40].

Thus, the data related to the recovery frequency of pathogens in the four experiments confirmed the presence of *Calonectria*, *Fusarium* and *Rhizoctonia* species in disease complex of milkwort. The frequency of *Calonectria* spp. and *Fusarium* spp. from infected tissues was very variable among the experiments and tested treatments whereas binucleate *Rhizoctonia* were averagely recovered with the lowest frequency among all three fungal genera.

## 3. Materials and Methods

### 3.1. Biological Control Agents and Fungicides

Four experiments were performed in a greenhouse within a nursery located in Carruba, Riposto, Catania province, Italy. In Experiments I and II, the efficacy of BCAs and fungicides, applied alone or in combination to control crown and root rot caused by artificial inoculations of *C*. *pauciramosa* on milkwort was evaluated. In Experiments III and IV, the efficacy of BCAs and fungicides, applied alone or in combination to control natural infections caused by *C*. *pauciramosa* on milkwort was evaluated. Moreover, when other pathogens affected milkwort, the effect on them were also evaluated. Fungicides and microbiological formulates tested are listed in the Table 5. Some of these already tested previously against *Calonectria* infections on other ornamental species [22,23] were chosen to confirm their activity on milkwort.

### 3.2. Effectiveness of Treatments

In all experiments (I, II, III and IV) performed under natural and artificial disease conditions, the effectiveness of treatments was always related to plant mortality values. In addition, it has been determined the recovery frequencies of involved fungal species associated to the decline and crown and root rot complex disease.

### 3.3. Experiment I

The experiment consists of 12 treatments replicated three times in a randomized complete block design (RCBD) with 50 to 70 cuttings of milkwort for replicate (Table 1). The same number of untreated cuttings served as control. Fungicide were applied 48 h (when in association with BCAs) and 24 h (when alone) before pathogen inoculation (with CP1) and BCAs were applied 24 h before inoculation. The treatment was performed with 100–140 mL of suspension for each replicate as a soil drench. Fungicides and BCAs were applied every 15 days. The crown area of each cutting was inoculated with approximately 0.5 mL of conidial suspension (10^5^ CFU) of *C*. *pauciramosa* in the same way. After inoculation, the cuttings were covered with plastic tunnel for 3 days and then maintained in a greenhouse at approximately 25 °C. The activity of fungicides and BCAs to control crown and root rot was evaluated 4 months after pathogen inoculation by calculating the plant mortality (%) and recovery frequency (%) of one pathogen species or more than one from symptomatic tissue. Plant mortality was calculated as percentages of cuttings death out of the total number of cuttings × 100. The recovery frequency was calculated as percentage of fragments from which colonies developed out of the total number of fragments examined (from 21 to 63 for replicate).

### 3.4. Experiment II

In this experiment, 10 treatments to control crown and root rot were tested (Table 2). Each replicate consisted of 72 cuttings of milkwort. Plant mortality, recovery frequency (on 14 fragments for replicate) and mean of number, length (cm) and weight (gr) of shoots on 50 plants for replicate were evaluated. The experiment was conducted as described above.

### 3.5. Experiment III

In this experiment, 9 treatments to control natural infections of milkwort were tested (Table 3). Each replicate consisted from 800 to 960 cuttings. Fungicides were applied every 15 whereas BCAs were applied every 30 days. Plant mortality, recovery frequency (on 70 fragments for replicate) and the mean of number, length (cm), weight (gr) and diameter of shoots (cm) on 50 plants for replicate were evaluated. The activity of fungicides and BCAs was evaluated after 5 months. The experiment was conducted as described above.

### 3.6. Experiment IV

In this experiment, 10 treatments to control natural infections of milkwort were tested (Table 4). Each replicate consisted of 1567 cuttings. Plant mortality was calculated as described above and recovery frequency (%) on 21–28 fragments for replicate were evaluated. The experiment was conducted as described above.

### 3.7. Recovery Frequency (%) and Identification of Isolates

Cuttings showing wilt and crown or root rot symptoms, were randomly collected from each treatment for evaluate the recovery frequency of pathogens. Fragments (each 5 × 5 mm) of symptomatic tissue (variable number depending on the experiment) were cut from the margins of lesions, surface-sterilized in a sodium hypochlorite solution (10%) for 20 s, followed by 70% ethanol for 30 s and rinsed three times in sterilized water. Tissue fragments were dried in sterilized filter paper, placed on 2% potato dextrose agar (PDA) amended with streptomycin (PDA) and were incubated at 25 °C. After 5–7 days, the different fungal species were counted. Subsequently, isolates single-conidium of each species recorded were identified. Characterization of fungal species was performed by molecular analysis. Anastomosis group (AG) of *Rhizoctonia* species was characterized by sequencing of the internal transcribed spacer region of rDNA and 5.8S (ITS) with primers ITS5 and ITS4 [41], whereas for *Fusarium* species was used the sequence of translation elongation factor 1-alpha (EF-1α) gene with primers EF1-728F and EF1-986R [42]. Genomic DNA was extracted using the Wizard Genomic DNA Purification Kit (Promega Corporation, WI, USA). The polymerase chain reaction (PCR) products were sequenced in both directions by Macrogen Inc. (South Korea). The DNA sequences generated were analyzed and consensus sequences were computed with Mega X [43]. BLAST searches were used to compare the sequences obtained with other sequences in the NCBI database. For the identification of *Calonectria* at species level, DNA sequencing of β-tubulin (benA), histone H3 (HIS3) and translation elongation factor-1α (TEF-1a) and a phylogenetic analysis of the multiloci concatenated genes was performed. Forty-five strains of *Calonectria* isolated from the cutting were characterized in this study, the sequences from 16 type strains of a closely related species were retrieved from a gene bank database (NCBI). Fungal growth and DNA extraction was performed as described previously in Vitale et al. [21]. Amplification of part of the β-tubulin gene (benA) was performed using the primers T1 [44] and CYLTUB1R [45]; for the Histone 3 region (HIS3) primers CYLH3F and CYLH3R [45] were used; and for the translation elongation factor-1α (TEF-1α) the primers EF1-728F [42] and CylEF-R2 [45] were used. Primers and relevant temperature used in the PCR analysis are the same used in Vitale et al. [21]. The preliminary alignment of the three sequenced loci (benA, HIS3, TEF-1α) was performed using the software package BioNumerics version 5.1 (Applied Maths) and manual adjustment for improvement was made by eye where necessary. The phylogenetic analysis was conducted firstly on the three single locus alignments and successively the combined alignment of the three loci was analyzed for inferring the organismal phylogeny. The multilocus alignment was conducted using the Clustal W algorithm in MEGA version X [43]. The best substitution model was calculated in MEGA software and then the evolutionary history was inferred by using the Maximum Likelihood method and the Hasegawa-Kishino-Yano model [46]. Initial tree(s) for the heuristic search were obtained automatically by applying Neighbor-Join and BioNJ algorithms to a matrix of pairwise distances estimated using the Maximum Composite Likelihood (MCL) approach and then selecting the topology with superior log likelihood value. A discrete Gamma distribution was used to model evolutionary rate differences among sites (5 categories (+G, parameter = 0.4736)). The rate variation model allowed for some sites to be evolutionarily invariable ([+I], 32.00% sites). The tree is drawn to scale, with branch lengths measured in the number of substitutions per site. This analysis involved 61 nucleotide sequences. All positions with less than 80% site coverage were eliminated, that is, fewer than 20% alignment gaps, missing data and ambiguous bases were allowed at any position (partial deletion option). There were a total of 1467 positions in the final dataset. Evolutionary analyses were conducted in MEGA version X [46].

### 3.8. Pathogenicity Test

The pathogenicity test was performed on potted healthy cuttings of milkwort inoculating the CP7 and CP14 isolates of *C*. *pseudomexicana*. For each isolate three replicates were used with 14 plants per replicate. Conidial suspensions of two isolates were obtained from 14-day-old colonies on carnation leaf agar (CLA) and inoculated at the crown of each plant. Un-inoculated plants served as control. After inoculation, plants were covered with a plastic bag for 48 h and maintained at 25 ± 1 °C and 95% relative humidity (RH) under a 12 h fluorescent light/dark regime until the symptoms were observed. Disease incidence (DI%) and plant mortality (PM%) was determined for each isolate 40 days after pathogen inoculation. Fungi were re-isolated from symptomatic tissues and identified as described above, to fulfil Koch’s postulates.

### 3.9. Statistical Analysis

Data about effectiveness of chemical, biological and combined treatments through four experiment were analysed by using the Statistica package software (version 10; Statsoft Inc., Tulsa, OK, USA). The arithmetic means of tested parameters were calculated, by averaging the values determined for the single replicates of each treatment. In the post-hoc analysis, the mean separation was conducted on all variables using post-hoc Fisher’s least significant difference test at α = 0.05. Prior to analysis, percentage values were transformed as arcsine square root (sin^−1^ square root x) to improve homogeneity of variances [47].

## 4. Conclusions

Our findings represent the first worldwide report about mixed infections caused by several fungal phytopathogenic genera, that is, *Calonectria, Fusarium* and *Rhizoctonia,* on milkwort in nursery although each of these fungi alone is well known as a causal agent of infections on stem, collar and root portions. Therefore, it should not be surprising as similar disease complexes will be detected in the future in other ornamental hosts. This paper clearly demonstrates that *C*. *pseudomexicana,* also involved into this disease complex, is already established in Italian ornamental nurseries probably imported on infected propagation material by trade exchanges from north Africa, where it was initially reported from other ornamental hosts [40]. Our findings also provide useful indications for management of these multiple infections in nursery. Comprehensively, boscalid + pyraclostrobin and fosetyl-Al (including their scheduled alternate application) partially confirmed their performances if compared to those previously reported for Calonectria infections alone [22,23]. Although BCAs provided averagely the lowest performances in managing this disease complex, they were able alone in most cases to reduce significantly multiple infections and sometimes when combined with fungicide to enhance the effects of active ingredients in nursery. Unlike previous research, this is mainly due to well-known factors in the differentiated effects exerted by single BCAs on one or the other pathogen involved in the disease complex.

## Figures and Tables

**Figure 1 plants-09-01682-f001:**
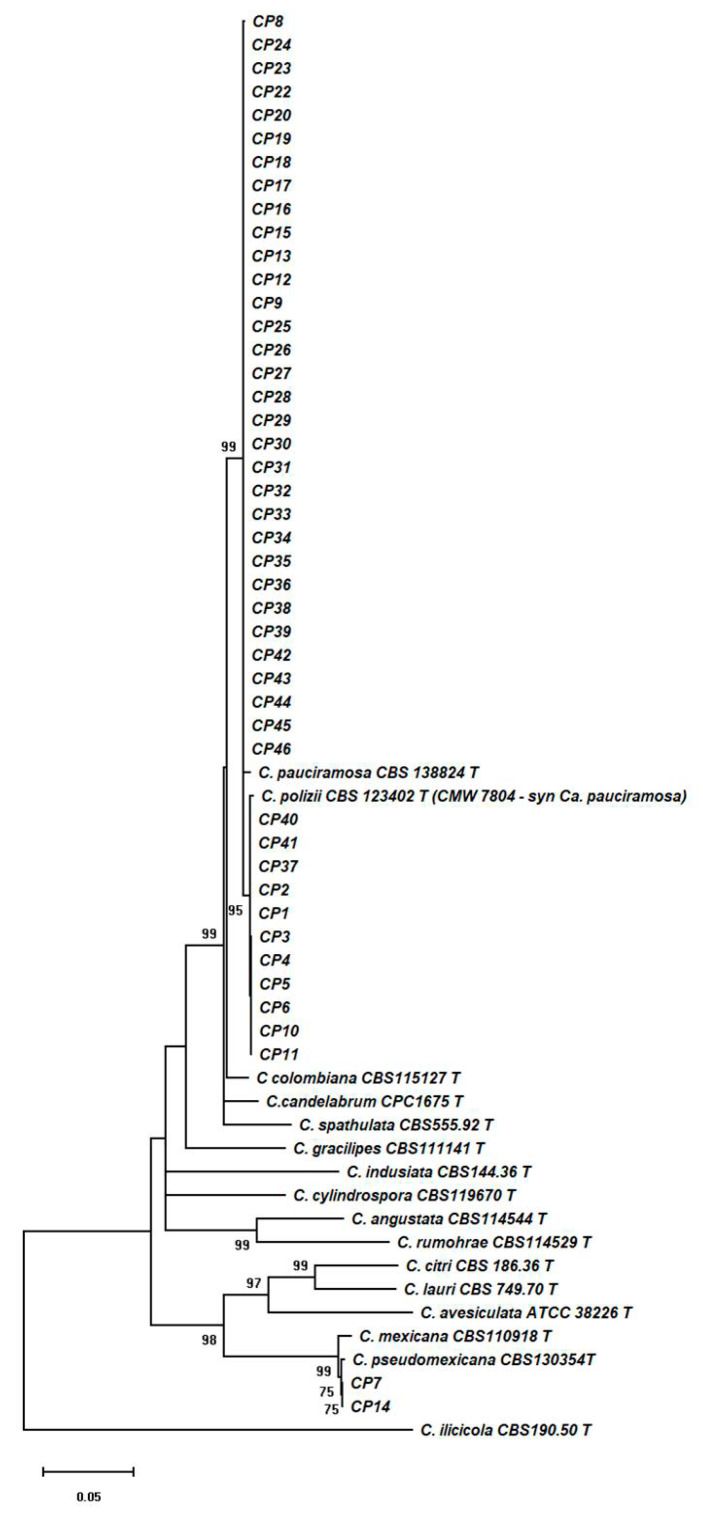
Multilocus (benA, TEF-1α and HIS-3) phylogenetic tree with the highest log likelihood (−7299.13) is shown. Numbers above branches are bootstrap values. Only values >70 % are indicated. CP acronym indicates *Calonectria* isolates collected and used for this study.

**Table 1 plants-09-01682-t001:** Efficacy of treatments in reducing milkwort decline in experiment I under artificial inoculation with *Calonectria pauciramosa* after 4 months.

Treatments	Plant ^z^	Recovery Frequency (%)	
	Mortality (%)	*Calonectria* spp.	*Fusarium* spp.
Fosetyl-Al	56.6 ± 1.6 e	14.6 ± 8.2 ^ns^	8.8 ± 3.2 ^ns^
Propamocarb & fosetyl-Al mixture	77.2 ± 3.8 bc	28.6 ± 14.3	9.5 ± 4.8
Boscalid + pyraclostrobin mixture	31.0 ± 2.6 f	4.2 ± 4.2	10.6 ± 0.6
*B. amyloliquefaciens* D747	71.1 ± 3.8 cd	11.1 ± 4.05	4.6 ± 2.0
*B. amyloliquefaciens* QST713	64.6 ± 1.5 de	19.5 ± 11.7	23.4 ± 7.7
*T. gamsii + T. asperellum*	65.8 ± 2.7 d	19.5 ± 5.5	17.9 ± 5.8
*Streptomyces* K61	80.1 ± 1.6 ab	1.2 ± 1.2	21.0 ± 5.8
*B. amyloliquefaciens* D747 & fosetyl-Al	81.4 ± 3.3 ab	30.2 ± 11.4	4.9 ± 3.7
*B. amyloliquefaciens* QST713 & fosetyl-Al	67.4 ± 2.3 d	42.9 ± 11.0	6.35 ± 4.2
*T. gamsii + T. asperellum* & fosetyl-Al	68.6 ± 3.3 d	31.1 ± 7.5	16.5 ± 8.3
*Streptomyces* K61 & fosetyl-Al	68.4 ± 2.4 d	26.8 ± 19.3	10.1 ± 10.1
Untreated control	84.9 ± 1.9 a	43.8 ± 7.40	27.1 ± 3.0

^z^ Mean data ± standard error of the mean (SEM) from three replicates are presented although analysis was performed on angular transformed values. Means followed by different letters within the column are significantly different according to Fisher’s least significance differences test (*α* = 0.05). ^ns^ = not significant data.

**Table 2 plants-09-01682-t002:** Efficacy of treatments in reducing milkwort decline and agronomic parameters of plants in experiment II under artificial inoculation with *Calonectria pauciramosa* after 4 months.

Treatments	Plant ^z^	Recovery Frequency *(%)*			Mean Shoot	Mean Shoot	Mean
	Mortality (%)	*Calonectria* spp.	*Fusarium* spp.	Binucleate *Rhizoctonia*	Number	Length (cm)	Weight (g)
Boscalid + pyraclostrobin mixture	45.1 ± 4.2 cd	19.0 ± 12.6 ^ns^	66.7 ± 33.3 ^ns^	0.0 ^ns^	5.7 ± 0.5 ^ns^	9.1 ± 0.7 ^ns^	80.8 ± 13.4 ^ns^
*T. harzianum* T22 and 908	77.2 ± 3.3 ab	45.2 ± 2.4	35.7 ± 14.3	21.4 ± 8.2	5.2 ± 1.0	9.0 ± 1.0	64.4 ± 8.4
*Streptomyces* K61	66.2 ± 6.5 bc	38.1 ± 21.2	50.0 ± 27.0	26.2 ± 14.5	5.4 ± 0.8	8.4 ± 0.7	60.6 ± 5.6
*T. atroviride* & *T. asperellum*	67.0 ± 15.9 bc	28.6 ± 7.1	83.3 ± 10.4	11.9 ± 11.9	4.9 ± 0.2	8.9 ± 0.3	65.8 ± 5.3
*B. amyloliquefaciens* QST713	69.1 ± 6.1 bc	33.3 ± 19.5	42.9 ± 18.9	19.0 ± 12.6	5.3 ± 0.2	9.2 ± 0.4	73.1 ± 6.6
Boscalid+pyraclostrobin & *T. harzianum* T22 and 908	47.6 ± 4.0 cd	21.4 ± 10.9	38.1 ± 8.6	2.4 ± 2.4	5.3 ± 0.5	8.0 ± 0.1	61.8 ± 4.1
Boscalid+pyraclostrobin & *Streptomyces* K61	54.0 ± 7.1 bcd	31.0 ± 11.9	61.9 ± 14.5	35.7 ± 12.4	6.3 ± 0.7	10.4 ± 0.6	92.2 ± 11.6
Boscalid+pyraclostrobin & *T. atroviride + T. asperellum*	48.0 ± 8.6 cd	19.0 ± 8.6	88.1 ±11.9	0.0	6.2 ± 0.4	9.6 ± 0.4	80.4 ± 4.2
Boscalid+pyraclostrobin & *B. amyloliquefaciens* QST713	39.7 ± 6.8 d	11.9 ± 6.3	59.5 ± 15.6	14.3 ± 14.3	6.0 ± 0.5	10.1 ± 0.8	84.5 ± 9.1
Untreated control	90.9 ± 4.7 a	45.2 ± 8.6	52.4 ± 9.5	11.9 ± 6.3	4.9 ± 0.1	7.8 ± 0.9	62.7 ± 8.9

^z^ Average data ± standard error of the mean (SEM) from three replicates are presented although analysis was performed on angular transformed values. Means followed by different letters within the column are significantly different according to Fisher’s least significance differences test (*α* = 0.05). ^ns^ = not significant data.

**Table 3 plants-09-01682-t003:** Efficacy of treatments in reducing milkwort decline and agronomic parameters of plants in experiment III under natural infection after 5 months.

Treatments	Plant ^y^	Recovery Frequency (%)			Mean Shoot	Mean Shoot	Mean	Mean
	Mortality (%)	*Calonectria* spp.	*Fusarium* spp.	Binucleate *Rhizoctonia*	Number	Length (cm)	Weight (g)	Diameter (cm)
Fosetyl-Al & *B. amyloliquefaciens* D747	7.1 ± 0.9 c	0.0 ^ns^	65.2 ± 6.9 ^ns^	8.3 ± 4.3 ^ns^	3.9 ± 0.1 ^ns^	13.3 ± 0.2 ^ns^	47.5± 2.1 ^ns^	0.4 ± 0.01 ^ns^
Fosetyl-Al & *T. gamsii + T. asperellum*	12.6 ± 3.0 bc	0.0	64.8 ± 2.1	9.0 ± 3.7	3.2 ± 0.2	16.6 ± 0.1	49.3 ± 1.4	0.4 ± 0.03
Fosetyl-Al & *T. asperellum* ICC012, T25, TV1	11.3 ± 3.3 bc	16.1 ± 8.2	60.9 ± 8.4	4.7 ± 1.3	2.4 ± 0.04	13.1 ± 0.1	37.0 ± 3.7	0.4 ± 0.02
Fosetyl-Al & *P. chlororaphis*	6.8 ± 0.7 c	5.2 ± 3.92	22.9 ± 2.2	6.7 ± 2.4	3.5 ± 0.2	14.2 ± 0.3	50.8 ± 3.6	0.4 ± 0.01
Fosetyl-Al & *Streptomyces* K61	7.2 ± 0.7 c	0.0 ± 0.0	52.5 ± 4.1	5.2 ± 1.3	3.2 ± 0.03	14.6 ± 0.3	42.3 ± 0.9	0.4 ± 0.01
Fosetyl-Al	13.9 ± 1.0 b	14.7 ± 11.2	38.8 ± 6.4	1.9 ± 0.5	2.6 ± 0.01	12.0 ± 0.5	33.0 ± 3.2	0.4 ± 0.02
Propamocarb + fosetyl-Al mixture	17.2 ± 2.6 ab	5.7 ± 3.00	44.3 ± 12.1	3.8 ± 2.1	2.8 ± 0.2	13.2 ± 0.5	34.7 ± 0.6	0.4 ± 0.02
Thiophanate-methyl & prochloraz	18.4 ± 4.2 ab	5.2 ± 5.2	17.6 ± 5.6	11.9 ± 4.2	2.4 ± 0.2	12.6 ± 0.1	37.0 ± 3.0	0.4 ± 0.02
Untreated control	20.6 ± 3.5 a	20.4 ± 10.3	72.0 ± 8.4	6.6 ± 3.3	2.8 ± 0.1	13.0 ± 0.8	35.8 ± 1.4	0.4 ± 0.01

^y^ Average data ± standard error of the mean (SEM) from three replicates are presented although analysis was performed on angular transformed values. Means followed by different letters within the column are significantly different according to Fisher’s least significance differences test (*α* = 0.05). ^ns^ = not significant data.

**Table 4 plants-09-01682-t004:** Efficacy of treatments in reducing milkwort decline in experiment IV under natural infection after 5 months.

Treatments	Plant ^z^	Recovery Frequency (%)		
	Mortality (%)	*Calonectria* spp.	*Fusarium* spp.	Binucleate *Rhizoctonia*
Boscalid + pyraclostrobin & fosethyl-Al ^(trts1)^	8.9 ± 0.4 ef	7.9 ± 4.2 ^ns^	86.9 ± 5.2 ^ns^	1.2 ± 1.2 ^ns^
Boscalid + pyraclostrobin & propamocarb + fosetyl-Al	15.9 ± 1.1 cd	22.2 ± 10.4	86.9 ± 4.3	1.2 ± 1.2
Scheduled trts1 & *B. amyloliquefaciens* D747	11.6 ± 0.5 ef	51.2 ± 4.3	79.8 ± 7.2	2.4 ± 2.4
Scheduled trts1 & *B. amyloliquefaciens* QST713	5.6 ± 0.3 g	34.9 ± 3.2	41.7 ± 6.3	3.6 ± 3.6
Scheduled trts1 & *Streptomyces* K61	8.7 ± 0.4 fg	11.5 ± 7.3	82.1± 10.9	1.2 ± 1.2
Scheduled trts1 & *T. gamsii + T. asperellum*	17.4 ± 0.7 bc	27.8 ± 14.3	52.4 ± 6.6	0
Scheduled trts1 *+ P. chlororaphis*	18.9 ± 0.9 bc	26.6 ± 4.6	79.8 ± 6.6	3.6 ± 2.1
Scheduled trts1 *+ T. harzianum* T22 and 908	12.4 ± 0.9 de	31.3 ± 10.6	57.1± 14.4	0
Untreated control	35.8 ± 3.1 a	54.4 ± 1.4	88.1 ± 6.6	1.2 ± 1.2
Standard trts (thiophanate-methyl & prochloraz)	21.9 ± 4.1 b	46.0 ± 5.7	19.1 ± 1.2	0

^z^ Average data ± standard error of the mean (SEM) from three replicates are presented although analysis was performed on angular transformed values. Means followed by different letters within the column are significantly different according to Fisher’s least significance differences test (*α* = 0.05). ^ns^ = not significant data.

**Table 5 plants-09-01682-t005:** Fungicides and biological control agents selected in the experiments.

Active ingredient	Trade Name	Manufacturer	Rates (g or mL/100 L)	Formulation ^x^	Experiment
Boscalid + pyraclostrobin mixture	Signum™	Basf Italia S.p.A.	100	26.7 + 6.7 WG	I, II, IV
Propamocarb + fosetyl-Al	Previcur Energy™	Bayer Crop Science S.r.l.	250	47.3 + 27.7 SL	I, III, IV
Fosetyl-Al	Aliette™	Bayer Crop Science S.r.l.	300	80 WG	I, III, IV
Thiophanate-methyl	Enovit Metil FL™	Sipcam Oxon S.p.A.	100	41.7 FL	III, IV
Prochloraz	Sportak 45 EW™	Basf Italia S.p.A.	100	39.8 EW	III, IV
*Bacillus amyloliquefaciens* subsp. *plantarum* strain D747	Amilo-X™	CBC (Europe) S.r.l.	250	25 WG	I, II, III
*Bacillus amyloliquefaciens* (formerly *B. subtilis*) strain QST713	Serenade Max™	Bayer Crop Science S.r.l.	400	15.67 WP	I, II, III
*Streptomyces* K61 (formerly *S. griseoviridis*)	Mycostop™	Danstar Ferment AG	25	33 WP	I, II, III, IV
*Trichoderma gamsii* (formerly *T. viride*) strain ICC080 + *T. asperellum* (formerly *T. harzianum*) strains ICC012, T25 and TV1	Radix Soil™	Isagro S.p.A.	250	2 + 2 WP	I, III, IV
*Trichoderma harzianum* strains T-22 and Item 908	Trianum-P™	Koppert B.V.	250	1.15 WP	II, IV
*T. atroviride* (formerly *T. harzianum*) strains T11 and IMI206040 + *T. asperellum* (formerly *T. harzianum*) strains ICC012, T25 and TV1	Tusal™	Newbiotechnic S.A.	300	0.5 + 0.5 WG	II
*Pseudomonas chlororaphis* strain MA312	Cedomon (Cerall™)	Koppert B.V.	400	9 AL	III, IV
*Trichoderma asperellum* (former *T. harzianum*) strains ICC012, T25 and TV1	Xedavir™	Xeda International S.A.	400	2.8 WP	III

^x^ Percentage of active ingredient; AL = any other liquid; WG = water dispersible granule; WP = wettable powder; SL = soluble concentration; SC = suspended concentration; EC = emulsifiable concentration.

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
