# Peer review of "Can Biological Control Agents Reduce Multiple Fungal Infections Causing Decline of Milkwort in Ornamental Nursery?"

_plants, 2020, doi:10.3390/plants9121682_

Round 1

Reviewer 1 Report

The manuscript " Can Biological Control Agents Reduce Multiple Fungal Infections Causing Decline of Milkwort in Ornamental Nursery?" investigates and discusses a relevant topic in the frame of chemical and biological control of fungal infestations on milkwort (Polygala myrtifolia).

As the authors state, the evaluation of biological control agents (BCAs) and fungicide alone are crucial elements on milkwort. The authors calculate the plant mortality and recovery frequency for biological control agents (BCAs) and fungicides. Information on potential effects biological control agents and fungicides on effects of disease might be useful to support the choice of the appropriate pesticide in the sense of biological or chemical control, if secured knowledge is transferred to the users in an appropriate way.

I believe that the topic of the study is appropriate for plants as it is. I didn’t detect methodological flaws and problems regarding the clarity of, and correspondence between, the results and the conclusions.

M&M

Add a section about statistical analysis

Why the authors did not perform chi square test and bonferroni analysis?

Tables

In Tables, number of the plants (n) from which the values were obtained should be given as separate column.

Author Response

Q. Add a section about statistical analysis. Why the authors did not perform chi square test and bonferroni analysis? In Tables, number of the plants (n) from which the values were obtained should be given as separate column.

A. I add a statistical analysis paragraph. In this paragraph it is explained a transformation of percent data in angular value - as arcsine square root (sin−1 square root x) - to improve the homogeneity of variances. The authors prefer not add additional column to not make too heavy the readability of tables

Reviewer 2 Report

Manuscript ID plants-997838, submitted to Plants, presents the results of several nursery trials aiming to evaluate the control of milkwort decline, as well as an investigation into the causal species of this disease. The scope of it appears suitable for this publication, under experimental plant science (phytopathology). It appears to be a new and original contribution, although there are some definitive similarities in the objectives and methodology used to https://doi.org/10.1094/PDIS-06-16-0801-RE, another study from three of the authors on this manuscript, on different host plants. I feel that this limits the originality of the work, although it remains relevant. I also feel that the investigation into causal species is of more interest that the control provided by conventional fungicides and biocontrol agents, although results of both are somewhat equivocal (which is not uncommon in the field of applied plant biology, when dealing with complex phenomena). The interpretations are generally sound and justified by the data. The manuscript is generally well presented and organized, and while the authors' meaning is mostly clear, the quality of the language is less than ideal (worse in some parts of the manuscript, eg the abstract) and the manuscript would benefit from a thorough proofreading by a native English speaker overall. "significant data" should be replaced by "statistically significant difference" throughout the manuscript, as applicable.

Specific comments and suggestions are outlined below.

Table 2 (Experiment I)
Plant mortality is high (over 30%) even with the most efficacious fungicide treatment (boscalid and pyraclostrobin). Possible that other pathogens (oomycetes, bacteria, viruses) are also present and contributing to the decline? Supported by the low recovery frequency for inoculated Calonectria spp. and Fusarium, as shown in table.

Table 3 (Experiment II)
Two of the biocontrol agent treatments, Streptomyces K61 and T. atroviride & T. asperellum, resulted in a statistically significant decrease of plant mortality on their own, and this was not statistically different from the combination of these biocontrol agents with Boscalid + pyraclostrobin. This is pretty remarkable efficacy for bioncotrol agents under high disease pressure, and worth highlighting in the results or discussion (especially given the proposed title of this manuscript).

Table 4 (Experiment III)
Under "Treatments", "P. chloraphis" should be changed to "P. clororaphis", as in Table 1.

Table 5 (Experiment IV)
Under "Treatments", "P. chloraphis" should be changed to "P. clororaphis", as in Table 1.

Author Response

Q1 Table 2 (Experiment I)
Plant mortality is high (over 30%) even with the most efficacious fungicide treatment (boscalid and pyraclostrobin). Possible that other pathogens (oomycetes, bacteria, viruses) are also present and contributing to the decline? Supported by the low recovery frequency for inoculated Calonectria spp. and Fusarium, as shown in table.

A1 The authors cannot exclude that week pathogens or other factors will contribute ex post to make heavier the symptoms by exploiting solely the attacks of three main fungal pathogens. Indeed the authors recovered only the three fungal pathogens at different frequency. As you state this is a complex phaenomen.

Q2 Table 3 (Experiment II)
Two of the biocontrol agent treatments, Streptomyces K61 and T. atroviride & T. asperellum, resulted in a statistically significant decrease of plant mortality on their own, and this was not statistically different from the combination of these biocontrol agents with Boscalid + pyraclostrobin. This is pretty remarkable efficacy for bioncotrol agents under high disease pressure, and worth highlighting in the results or discussion (especially given the proposed title of this manuscript).

A2 Thank you for the comment. I implement the relative results in the manuscript.

A3 Table 4 (Experiment III)
Under "Treatments", "P. chloraphis" should be changed to "P. clororaphis", as in Table 1.Table 5 (Experiment IV)
Under "Treatments", "P. chloraphis" should be changed to "P. clororaphis", as in Table 1.

Q3. Thank you for the comment. Done